# 3D Printing of Thermo-Sensitive Drugs

**DOI:** 10.3390/pharmaceutics13091524

**Published:** 2021-09-21

**Authors:** Sadikalmahdi Abdella, Souha H. Youssef, Franklin Afinjuomo, Yunmei Song, Paris Fouladian, Richard Upton, Sanjay Garg

**Affiliations:** 1Pharmaceutical Innovation and Development (PIDG) Group, Clinical and Health Sciences, University of South Australia, Adelaide, SA 5000, Australia; sadikalmahdi.abdella@mymail.unisa.edu.au (S.A.); souha.youssef@mymail.unisa.edu.au (S.H.Y.); Franklin.Afinjuomo@unisa.edu.au (F.A.); May.Song@unisa.edu.au (Y.S.); paris.fouladian@mymail.unisa.edu.au (P.F.); 2Department of Pharmacology and Clinical Pharmacy, College of Health Sciences, Addis Ababa University, Zambia St., Addis Ababa 1000, Ethiopia; 3Clinical and Health Sciences, University of South Australia, Adelaide, SA 5000, Australia; richard.upton@unisa.edu.au

**Keywords:** 3D printing, thermolabile, thermosensitive, FDM, SSE, SLA, DLP

## Abstract

Three-dimensional (3D) printing is among the rapidly evolving technologies with applications in many sectors. The pharmaceutical industry is no exception, and the approval of the first 3D-printed tablet (Spiratam^®^) marked a revolution in the field. Several studies reported the fabrication of different dosage forms using a range of 3D printing techniques. Thermosensitive drugs compose a considerable segment of available medications in the market requiring strict temperature control during processing to ensure their efficacy and safety. Heating involved in some of the 3D printing technologies raises concerns regarding the feasibility of the techniques for printing thermolabile drugs. Studies reported that semi-solid extrusion (SSE) is the commonly used printing technique to fabricate thermosensitive drugs. Digital light processing (DLP), binder jetting (BJ), and stereolithography (SLA) can also be used for the fabrication of thermosensitive drugs as they do not involve heating elements. Nonetheless, degradation of some drugs by light source used in the techniques was reported. Interestingly, fused deposition modelling (FDM) coupled with filling techniques offered protection against thermal degradation. Concepts such as selection of low melting point polymers, adjustment of printing parameters, and coupling of more than one printing technique were exploited in printing thermosensitive drugs. This systematic review presents challenges, 3DP procedures, and future directions of 3D printing of thermo-sensitive formulations.

## 1. Introduction

Three-dimensional printing (3DP), also known as additive manufacturing, rapid prototyping, and solid freeform fabrication, enables the construction of bespoke objects in a layer-by-layer fashion [1,2]. It encompasses a multitude of different technologies classified into seven main categories: binder jetting, directed energy deposition, material extrusion, material jetting, powder bed fusion, sheet lamination, and vat photopolymerisation [3]. Interestingly, this innovative technology has established its roots in various disciplines ranging from arts and engineering to implants and regenerative medicine.

The introduction of 3DP dates back to 1981 when Dr. Kodama attempted to prepare 3D-shaped objects by exposing photosensitive resins to ultraviolet (UV) rays using a single laser beam controlled by a mask [4]. Although his work was not patented at the time, it initiated a tight race towards developing 3DP techniques. Kodama’s concept later led to the development of stereolithography (SLA) technology. The SLA was taken up by Charles Hull who patented it in 1986. The process involved hardening liquid polymers layer by layer using UV light, and the procedure was orchestrated by computer-controlled beam and digital data. Shortly after, 3D systems released the first 3D commercial printer in the world (SLA-1) [5]. Subsequently, selective laser sintering (SLS) and fused deposition modelling (FDM) were patented in 1988 and 1992, respectively. SLS is based on fusing powdered printing materials by laser beams, whereas FDM involves depositing layers of molten polymers extruded from a heated nozzle. The nozzle movement is controlled by a computer executing a designed digital model [6,7]. The binder jet printer (BJ), also called powder bed and inkjet head 3DP, was developed and patented by Sachs et al. in 1993 [8]. Afterward, Thomas Boland created the first bioprinter that uses bio-inks to fabricate different objects including biological tissues using a semi-solid extrusion mechanism [9] (Figure 1).

The conventional drug treatment approach is often based on “one size fits all” where most patients receive the same drugs at the same doses and frequencies, which has shown varied responses and has been linked to adverse drug reactions [10]. Personalised medicine is becoming very popular in recent years and refers to an approach of tailoring medical treatment to patients on the basis of their characteristics such as genetic profile, concurrent medicines, and disease state. Personalised medicine is safer and more efficacious, improves patient compliance, and is cost-effective [11]. Nevertheless, the progress of personalised medicine has been partially limited by the absence of appropriate dosage forms of desired dose strength, shape, and colour, among others. Pharmaceutical industries develop drugs according to the most abundant and representative therapeutic response profile. It does not take all particularities of every individual into account. This is particularly evident for paediatric and geriatric populations, increasing the risk of treatment failure or adverse effects. Dose adjustment according to weight, age, and pharmacogenetic and pharmacokinetic characteristics are key to achieving the desired therapeutic effect and improving the efficacy/toxicity balance. Likewise, a modification of colours, flavours, and even the form of the solid dosage forms would considerably increase the adherence to treatment in children and the elderly [12]. 3DP has the potential to play a revolutionary role to fill the current gap associated with conventional drug production. It fabricates unlimited 3D dosage forms designed using computer-aided design (CAD), offering the possibility for tailored drugs and personalised treatment options.

3DP techniques were first introduced into the pharmaceutical sector in the early 1990s when Sachs et al. invented and patented a rapid prototyping method entitled “three-dimensional printing techniques” at Massachusetts Institute of Technology (MIT, Cambridge, MA, USA). 3DP represented a major revolution in the pharmaceutical industry for the fabrication of 3D medical products and drug formulations in the pharmaceutical industry [13,14]. Among the many 3D techniques, only five of them, namely BJ, FDM, SLS, SLA, and semi-solid extrusion (SSE), have been explored for pharmaceutical application [15]. The U.S. Food and Drug Administration (FDA) approval of the first 3D-printed tablet, Spritam^®^ (levetiracetam), in August 2015 started a new era in pharmaceutical production and increased the interest in the technology for the production of various dosage forms [16,17].

### 1.1. Advantages of Three-Dimensional Printing (3DP) in Dosage Form Production

Mass/bulk production of pharmaceuticals has been the common practice over the years, but it does not allow for much flexibility to personalise therapy. As we launch into the future, all aspects of life have evolved, and medical care is no exception. 3DP offers several advantages such as the potential to develop pharmaceuticals with improved efficiency, compliance, and individuality. The advantages of 3DP are discussed below (Figure 2).

#### 1.1.1. Customisation

There are several potential advantages of customised drug dose forms. First, the demand for tailored custom formulations has significantly increased with the increase in knowledge and advancement of pharmacogenomics. 3DP allows for the production of dosage forms with precise dosing for different patient groups, including those with pharmacogenetic polymorphism [18]. Second, one or more excipients can be changed or removed for patients with allergies to a specific ingredient. Finally, taste can be masked and swallowability can be significantly improved for paediatrics and patients with dysphagia [19]. Several studies have reported the design and production of dosage forms with precise doses and fast disintegrating tablets using 3DP. For example, Wang and his colleagues fabricated a taste-masked donut-shaped tablet by retarding dissolution in the oral cavity [20].

#### 1.1.2. Polypharmacy

The administration of multiple drugs simultaneously has been a topic of interest in the medical field and illustrated to be advantageous for treating complex disease conditions such as hypertension, tuberculosis, HIV, and diabetes [21,22,23,24]. Despite the advantages, patients on polypharmacy are at higher risk of drug interactions and side effects [23]. Moreover, poor adherence to medications has also been well-documented among patients taking multiple medications. Simplification of drug therapy based on the intake of a single pill containing all the drugs (polypill) would improve adherence to the prescribed treatment and probably enhance the efficacy/toxicity balance. Several studies have reported the fabrication of customised polypills using 3DP [25,26,27].

#### 1.1.3. Safety

Drugs with a narrow therapeutic window such as warfarin should be given in a precise dose to avoid adverse drug reactions. Furthermore, there are several drugs that require dose titrations based on patient response. Splitting tablets manually or dissolving in different solutions have been linked to undesirable outcomes [28]. 3DP allows for the production of precise doses of these drugs and minimisation of the risk of adverse effects. According to the Food and Drug Administration (FDA), drugs with a narrow therapeutic index require careful titration for safe and effective use. Unique precise dosing of these drugs can be achieved at the decentralised point of care, avoiding splitting of tablets to adjust dose [29].

#### 1.1.4. Flexible Design

Regardless of the type of printing technique, the process of 3DP starts with a computer-aided design (CAD). The movement of the printing head is digitally controlled and allows for the production of dosage forms of any shape or size with varying complexities. 3DP has been used to develop formulations with appealing shapes for children [30], tablets of different shapes for easier swallowing [31], and customised drug loaded stents for cancer treatment [32]. The flexibility of design also provides control over drug release. A study has shown that various shapes of tablets have different rates and consequently different drug release rates. Surface area/volume ratio (SA/V) was found to be the main determinant, wherein SA/V was directly proportional to the drug release rate [33].

#### 1.1.5. Point of Care Production

3D printers are portable, economical, and relatively simple to operate, which makes them suitable for manufacturing decentralised dosage forms at the point of care. It is well-situated to address supply–demand imbalances of medications that occur in emergencies such as the unprecedented pandemic of SARS-COVID-2 or in resource-constrained settings such as disaster areas, emergency departments, first response units, and military operations. Moreover, unstable drugs with a short half-life can be produced using 3DP on demand [17,34].

#### 1.1.6. Waste Minimisation

3DP is considered eco-friendly due to it having lower material consumption as the process utilises about 90% of starting materials. Once the product is printed out, no extra parts need to be removed, leading to less waste [35].

### 1.2. Challenges and Methods of Printing Thermolabile Drugs

Temperature-sensitive pharmaceutical products are generally defined as “any pharmaceutical good or product which, when not stored or transported within predefined environmental conditions and/or within predefined time limits, is degraded to the extent that it no longer performs as originally intended” [36]. Whilst the definition is commonly applied to drugs that must be stored in a refrigerator, it can also apply to drugs that can be degraded in the printing process with high printing temperature or UV light/beams. Many drugs including biological agents can be easily degraded and lose their activity at the high temperatures applied in commonly used printers, revealing the need for a suitable printing platform for these compounds. The fact that more thermolabile drugs are becoming available and expensive, aside from the increased popularity of 3DP technology to fabricate personalised doses, warrants significant attention [37].

Among the commonly used 3DP techniques, FDM has been explored, widely owing to its printing precision and low cost. FDM involves the preparation of optimised filaments by melting active pharmaceutical ingredients (API) and pharmaceutical-grade polymers using hot-melt extrusion. Subsequently, the filaments are heated and extruded through a nozzle tip, followed by layer-by-layer deposition and solidification onto a build plate into the desired geometry (Figure 3a). FDM has been proven to offer a wide range of advantages such as dispensing of a precise dose by varying the tablet size [38,39] or filament drug load [40], influencing drug release by printing different geometries and dual extrusion FDM for a combining different drugs or polymers in one dosage form [41,42]. FDM is the most widely evaluated 3D printing technique in the pharmaceutical sector, owing to the use of a relatively straightforward process and less expensive equipment, a diverse choice of excipients, and ease of producing dosage forms. Many dosage forms including tablets, polypills, controlled-release devices, and oro-mucosal films have been fabricated using this technique [43,44,45,46]. The main drawback of FDM is the possible risk of drug degradation that may result from the use of a significant amount of heat. Indeed, in many cases, a temperature higher than 120 °C was used, which can lead to drug degradation, deterioration of mechanical properties, a decrease of the physical stability, filament aging, and relatively poor resolution of the 3DP objects [47]. Moreover, the bioactivity of the drugs can be altered due to the high melting temperature required to extrude the filaments, thus rendering the FDM technique incompatible with APIs that are thermolabile [48].

SLA is another 3DP technique commonly used for the preparation of pharmaceutical dosage forms. It involves the incorporation of the drug into a photo-curable polymer and the addition of a photo-initiator (PI); the resulting gel is then exposed to highly controlled UV beams which solidify the polymer layer by layer until the product is obtained [49] (Figure 3b). No heat is required during the process, reducing the chance of drug degradation, a property that can be considered useful for the printing of tablets with thermo-sensitive drugs. Nevertheless, the use of UV light for extra post-curing steps to finalise the product increases the risk of drug and excipient degradation. The availability of a limited number of photocurable polymers and the potential toxicity of PI also limits the use of SLA for printing drugs in general and thermolabile drugs in particular [50]. SLS, on the other hand, uses a powder bed as printing material and laser beams for particle binding (Figure 3c). SLS offers several advantages such as the high resolution of the finished product, improved reproducibility, solvent-free procedure, and ease of preparation. Nonetheless, drug degradation from high energy input of the laser beams has been the main concern [49]. SSE, also known as pressure-assisted microsyringe (PAM) printing, is a recently introduced technique that involves depositing a gel or a paste through a syringe attached to the printing head, one layer at a time (Figure 3d) [51]. The right gel/paste consistency can be achieved by heating or mixing the feedstock with a solvent or mixture of solvents [52]. The material hardens, upon extrusion, allowing the subsequent tiers to be supported by the ones underneath [53]. In contrast to other printing techniques such as FDM, SSE uses semi-solid or semi-molten material as a starting material to produce the desired dosage form [54]. SSE, also known as bioprinting, has been extensively used in tissue engineering, but its use in drug development remained suboptimal. The technique attracted huge attention for the formulation of thermolabile drugs as printing is carried out under mild conditions. Moreover, SSE is considered a simpler procedure where ingredients can directly be added to the gel-like feeding material without additional steps [55]. It is worth noting that the issue of getting a suitable gel with the right viscosity for printing has been a great challenge with this printing technique.

Several attempts have been made over the past few years to reduce the risk of drug degradation using existing 3D printing technologies. Despite many efforts to modify the printing techniques to fabricate thermolabile drugs, the strategies have not been addressed comprehensively. To our knowledge, this is the first review paper addressing the printing of thermolabile drugs using 3D printing technology. This review aims to address different strategies used and modifications made to existing processes to print thermolabile drugs. This review provides deep insight and a better understanding regarding the 3D printing of thermolabile drugs.

## 2. Methods

This review was based on the Preferred Reporting Items for Systematic Reviews and Meta-Analyses (PRISMA) guidelines. Electronic databases (Web of Science and PubMed) were used to survey the literature. The keywords selected for this review were divided into two parts. The first part comprised terms mostly used to describe the technology: “3D printing”, “three-dimensional printing”, “additive manufacturing”, “bioprinting”, “pressure-assisted microsyringe”, and “semi-solid extrusion printing”. Terms used to describe heat-sensitive drugs such as “thermolabile”, “thermosensitive”, heat-sensitive”, and “heat-labile” constituted the second part. Furthermore, the results from publications related to 3DP were limited to pharmaceutical application by excluding articles in other areas. This was required as the 3DP has a variety of applications in many different fields.

The search phrase used in Web of Science was “(TOPIC: (three dimensional printing) OR TOPIC: (3D printing) OR TOPIC: (bioprinting) OR TOPIC: (semi-solid extrusion printing)) AND (TOPIC: (thermo-labile drug/s) OR TOPIC: (thermo-sensitive drug*) OR TOPIC: (heat-labile drug*) OR TOPIC: (heat-sensitive drug*))” and in PubMed was “((((thermo-labile drug*) OR thermo-sensitive drug*) OR heat-labile drug*) OR heat-sensitive drug*)) AND ((((3D printing) OR three dimensional printing) OR bioprinting) OR semi-solid extrusion printing).” Additional searches were also conducted using the terms “fused deposition modeling” or “FDM”, “stereolithography” or “SLA” and digital light processing or “DLP” to identify techniques relevant to fabrication of thermosensitive drugs, and all relevant papers were included in the review. In this review, a selection of criteria was designated for determining which articles were to be included. These were (1) the technology should be 3D printing, (2) the drugs should be heat-sensitive, and (3) the printed objects should be drugs. Moreover, only original research articles were included, and other publications such as conference papers and review articles were excluded.

## 3. Results and Discussion

### 3.1. Literature Search Output

The search engines resulted in a total of 822 articles. The digital object identifier system was used to remove duplicates, which resulted in 167 articles. We performed study selection using Covidence in a two-step process, where two review authors (S.H.A. and S.H.Y.) independently screened all titles and abstracts on the basis of the inclusion criteria and subsequently retrieved all relevant data on the basis of the criteria mentioned in the Section 2. Any discrepancies were resolved by discussion until consensus was reached. The excluded articles totalled 207, leaving 81 articles to be included in this review (Figure 4).

### 3.2. Printing Techniques and Extrusion Temperature

Several 3DP techniques were reported for the printing thermolabile drugs. Commercial and built-in-house printers were employed for the fabrication of the dosage forms. SSE was the most used 3DP technique and applied in 23% (*n* = 19) of the studies considered in this systematic review. FDM, SLA, and DLP were applied in 15% (*n* = 12), 14% (*n* = 11), and 12% (*n* = 10) studies, respectively. FDM coupled with filling (automatic/manual) was mentioned in 23% (*n* = 18) of the studies. A combination of other techniques such as FDM and binder jetting, FDM and super critical fluid technology, and SLA and inkjet were also reported in the literature. All studies used commercial printers except two studies that used an in-house-developed printer to fabricate drug dosage forms (Figure 5, Table 1).

#### 3.2.1. Semi-Solid Extrusion

Several studies have demonstrated the suitability of SSE for the production of thermosensitive drugs. For instance, Dores and his colleagues reported the production of theophylline tablets at a moderate temperature range (65–100 °C) using SSE extrusion printing. The temperature was lower than that commonly used in FDM and other printing techniques, making the process suitable for moderately thermolabile drugs. The tablet was produced using a hybrid approach that uses an extrusion-based system delivered by a simple metal syringe. The pharmaceutical ink was prepared using different grades of poly(vinyl alcohol) (PVA) and polyvinylpyrrolidone (PVP), together with plasticiser and lubricants. The use of water as a temporary plasticiser improved material flow and enabled printing at a lower temperature. Sorbitol, lactose, and D-mannitol were examined as fillers to assess the compatibility of the process with different fillers, and all of them yielded well-structured tablets, indicating the versatility of the method. The material flow was also further enhanced by the addition of sodium stearyl fumarate 5% as a lubricant [56]. Likewise, semi-solid extrusion (SSE) was used to produce solid lipid tablets incorporating a poorly water-soluble drug, fenofibrate. Fenofibrate tablets were successfully printed from emulsion gels at room temperature, making the methodology particularly useful for some thermolabile compounds. The hydrogel was prepared in a two-step process. First, an emulsion was prepared by adding the required amount of drug-loaded lipid-based-formulation (LBF) (oil phase) to Milli-Q water, followed by a two-step emulsification process. Subsequently, a suitable hydrogel was prepared by mixing oil in water (O/W) emulsion with methylcellulose and croscarmellose sodium at 70 °C. Emulsion gels were then printed into tablets using a BIO X 3D- printer equipped with a pneumatic printhead (Cellink, Gothenburg, Sweden). The printing was performed at room temperature, illustrating the potential use of the method for printing thermosensitive drugs (Figure 6) [57].

Kuźmińska and co-workers reported an innovative solvent-free direct extrusion 3D printing process that operates at a moderate temperature range (90–110 °C). The process was not only performed at lower temperatures but also eliminated the post drying step that may degrade some drugs. The feed was prepared by mixing methacrylate polymer(s) (Eudragit RL and RS), glycerol monostearate (GMS), and theophylline using a mortar and pestle. The blends were then transferred to a grinder where it was shear-mixed with Triethyl citrate (TEC). The incorporation of fatty glyceride (GMS) demonstrated a dual temperature-dependent behaviour by acting as a plasticiser and a lubricant at the printing temperature while aiding solidification at room temperature. Finally, the blends (approximately 10 g) were filled in a 12 mm diameter metal syringe (Hyrel 3D, Atlanta, GA, USA) to produce the tablet (Figure 7). The work revealed a simplified, facile, and low-cost 3D printing for small-batch manufacturing of bespoke tablets that circumvents the use of high temperature and post-manufacturing drying [58].

In a related study, SSE was also used to prepare immediate-release tablets at room temperature, reducing the risk of drug degradation. Levetiracetam (LEV) tablets, an anti-epileptic drug that requires frequent dose titration of various volumes, were successfully printed using a commercial SSE printer (MAMII; Fochif Mechatronics Technology Company, Ltd., Shanghai, China). The LEV paste was prepared by first mixing levetiracetam powder and polymers (sodium carboxymethyl cellulose, PVP K30, and carboxymethylcellulose sodium) in a mortar and pestle, which was subsequently added to a certain volume of pre-mixed ethanol and water. LEV tablets equipped with a smooth appearance and excellent mechanical properties were successfully printed at room temperature [59]. Gastro-floating tablet of dipyridamole, a drug with poor water-solubility and short biological half-life, was another example of dosage form prepared using the SSE printing technique at room temperature. Firstly, hydroxypropyl methylcellulose (HPMC) gel (1%, *w*/*v*) was prepared by dissolving HPMC E15 in water. The dipyridamole fine powder and other required excipients were mixed for 30 min. Then, HPMC E15 (1%, *w*/*v*) gel was mixed with 95% ethanol at a fixed ratio to form a hydro-alcoholic gel. Finally, the obtained hydro-alcoholic gel was added into the powder and mixed until a homogenous paste without aggregates or separation was achieved. The tablets with different infill architectures were printed using a commercial extrusion-based printer at room temperature (Figure 8) [60].

SSE extrusion-based printing technique was also explored to fabricate novel gastro-floating tablets. For instance, Real and co-workers demonstrated the preparation of a floating sustained release tablet using an innovative melting solidification printing process (MESO-PP), a semi-solid extrusion technique, avoiding the use of solvents and high temperatures. A fixed ratio of Gelucire^®^50/13 (fatty polyethylene glycol esters) and ricobendazole (RBZ) were melted at 60 °C under continuous stirring. The mixture was then poured into a syringe and the tablets were printed using a commercial printer (3-Donor^®^developed by Life SI) at a printing temperature of 49 °C [61]. Moreover, Li and co-workers investigated the development of a novel puerarin gastric floating system with a concentric annular internal pattern using a 3D extrusion-based printing technique. A uniform and smooth puerarin paste was prepared by mixing all ingredients with a given amount of ethanol aqueous solution. The tablets were then fabricated from the prepared paste using a commercial microextrusion 3D printer (Fochif Mechatronics Technology Co., Ltd., Shanghai, China) at room temperature [62]. Likewise, a commercial pressure-assisted syringes 3D printer (Fochif Mechatronics Technology Co., Ltd., Shanghai, China) was employed to produce gastro-retentive drug delivery systems loaded with ginkgolide at a printing temperature of 25 °C [63].

Interestingly, SSE extrusion has also been used to fabricate blank and drug-loaded drug-eluting constructs/scaffolds. Naseri et al. reported a novel low-temperature (20 °C) 3D printing technique based on SSE poly-lactic-co-glycolic acid constructs using methyl ethyl ketone (MEK) as a solvent. The solvent was finally removed following printing. The drug-eluting constructs were suggested to be a promising platform to incorporate thermolabile drugs [64]. In the same fashion, a rifampicin-loaded 3D scaffold was prepared using SSE 3D printing for the treatment of osteomyelitis. A biodegradable polymer, polycaprolactone (PCL), was used to load heat-labile antibiotic, rifampicin. The scaffold was successfully printed at a printing temperature of 60 °C and the drug was stable after printing (Figure 9) [65]. Drug-eluting polycaprolactone/nano-hydroxylapatite (PCL/nHA) nanocomposites loaded with vancomycin and ceftazidime was also fabricated using built-in-house solution-extrusion printer, illustrating the possibility of using SSE-based printing technique to develop implants for different medical applications [66]. It is worth noting that coaxial SSE 3DP, where the coaxial extruder fixed to the printhead of the Ultimaker3 printer was connected through tubes to the syringe pump system, was used for the fabrication of propranolol-loaded drug delivery system [67].

Furthermore, several other studies reported the use of SSE printing techniques to fabricate tablets [51,68,69,70,71], oral films [72], and scaffolds [73], illustrating that SSE is an excellent technique to fabricate thermosensitive drugs at a lower temperature.

#### 3.2.2. Fused Deposition Modelling

Several efforts have been made thus far to utilise FDM to fabricate thermolabile drugs. For instance, Okwuosa and co-workers used a lower-temperature FDM to produce immediate-release tablets of two model drugs (theophylline and dipyridamole). The drug-loaded filament was prepared by adding a mixture of a polymer (PVP), plasticiser (TEC), filler(talc), and API (theophylline or dipyridamole) into the HME heated to 100 °C to allow for homogenous distribution. For the printing of tablets, pre-prepared filaments were fed into a commercial FDM 3D printer (Makerbot Industries, Brooklyn, NY, USA, USA) and printed at a relatively lower temperature (110 °C). The tablets showed excellent mechanical properties and acceptable in-batch variability [74]. In a similar work, Kempin et al. reported the use of polymers with a low glass transition temperature (Tg) to fabricate tablets of thermosensitive drugs using FDM. A pantoprazole tablet composed of polyethylene glycol (PEG 6000) (loaded with 5% and 10% of pantoprazole) was successfully printed at a nozzle temperature of 54–55 °C. Moreover, a higher molecular weight PEG 20000 loaded with 10% pantoprazole was printed at 60 °C. Pantoprazole-loaded (10 to 30%) tablet prepared from PVP and TEC was printed at a relatively higher temperature in the range of 79–87 °C. Poloxamer 407 and Kollidon VA64 were also used to prepare a tablet of pantoprazole at a temperature below 100 °C, highlighting the potential use of the technique for thermolabile drugs [75] (Figure 10). Other studies [76,77] used PCL to fabricate implants using FDM printers at moderately low temperatures. Kollamaram and his colleagues explored polymers that print at lower temperatures. KollidonVA64 alone or in combination with Kollidon 12PF was used to successfully manufacture printlets (printed tablets) of ramipril (melting point: 109 °C) using FDM, MakerBot Replicator2X Desktop (MakerBot Industries., Brooklyn, NY, USA) printer, at an extrusion temperature of 90 °C with no signs of degradation, illustrating the role of selecting the right polymers which can help in reducing printing temperature, thereby reducing the risk of drug degradation. First, a filament containing Kollidon VA 64, PEG 1500, mannitol, ramipril, and magnesium carbonate was extruded using a single-screw extruder (Noztek Profilament extruder, NozteK, Shoreham-by-Sea, UK) at 70 °C, and subsequently, a tablet containing ramipril was fabricated [78]. Paracetamol tablets were also successfully produced at a moderate printing temperature of 90 °C using a mixture of potato starch and hydroxypropyl cellulose by employing an FDM printer [79]. Katsiotis et al. demonstrated an interesting concept of increasing thermal stability of drugs using mesoporous magnesium carbonate, which may be useful for protecting thermosensitive drugs against heating applied in FDM [80].

Patel and co-workers applied a novel acid-based super solubilisation (ABS) principle to reduce printing temperature while preparing a tablet of haloperidol using FDM printing. The acid–base interaction between glutaric acid and haloperidol in the present investigation led to the formation of amorphous and viscous materials, which reduced the complex viscosity of formulations as functions of temperature and greatly improved their melt extrudability into filaments and printability of filaments into tablets. The interaction helped to print the tablet at relatively low printing temperature (115–120 °C), illustrating that the principle can be used for thermosensitive drugs [81].

#### 3.2.3. Stereolithography and Digital Light Processing

Both SLA and DLP rely on polymerisation and solidification of photocurable resins upon exposure to a light source [82]. As the process requires no heating, it could be considered for printing thermo-sensitive drugs. Several studies reported the use of SLA and DLP for the fabrication of drug dosage forms. For example, paracetamol-containing tablets were printed by SLA printers using PEGDA as a common photoreactive polymer at room temperature [33]. SLA printing technique was also used to fabricate drug-loaded hydrogels, providing novel manufacturing protocols for vitamin C and other water-soluble vitamins [83] and ibuprofen [84]. Polypill containing irbesartan, atenolol, hydrochlorothiazide, and amlodipine was successfully fabricating using SLA at room temperature, illustrating the flexibility of SLA to fabricate different dosage forms [85].

Additionally, SLA was used to fabricate drug-loaded implants [86], scaffolds [87], polypills [25], and tablets [18]. It is noteworthy that some studies reported undesirable reactions between drugs and damage to some drugs in the printing process [85,86] that requires consideration during fabrication of drugs, particularly thermosensitive drugs.

Similarly, DLP was used to fabricate several dosage forms. The first oral dosage form using this technique was reported using poly(ethylene glycol) diacrylate (PEGDA) and poly(ethylene glycol) dimethacrylate (PEGDMA) serving as photoreactive polymers, 2-hydroxy-4′-(2-hydroxyethoxy)-2-methylpropiophenone as a photoinitiator, and theophylline as a model drug [88]. Other PEGDA-based tablets loaded with different concentrations (5–20%, *w*/*w*) of acetaminophen, theophylline, and carbamazepine separately were also reported. The printing ink was prepared by mixing PEGDA, PEG400, diphenyl (2,4,6-trimethyl benzoyl) phosphine oxide (photoinitiator), drug, and water and mixed at room temperature. The study investigated the effect of tablet composition and printing conditions on the drug’s release [89].

Additionally, atomoxetine-containing tablets using DLP were developed in two different research articles. One utilised the tablets to develop artificial neural networks (ANN) predictive models where the release rates were studied for tablets with different thicknesses and drug loading [90], whereas the other study evaluated how the formulation composition affected atomoxetine release and kinetics, as well as the mechanical properties of the tablets [91]. Likewise, paracetamol was incorporated in tablets, and its release, tensile strength of tablets, dissolution rate, and internal structure were assessed upon varying tablet ingredients [92]. Madzarevic et al. utilised the same concept to develop ibuprofen-loaded tablets to investigate the effect of formulation factors on the printability of the tablet and predict the extended-release, making use of the ANN model [93].

DLP was also used to fabricate drug-loaded implants [94], moulds [95], and microneedles [96,97], among others, highlighting its versatility to produce different dosage forms at mild conditions.

#### 3.2.4. Combination of Technologies

##### FDM and Filling Techniques

Several studies reported the coupling of FDM with other techniques. The coupling of the techniques has avoided the exposure of the drugs to the high temperature used in FDM. For instance, Petra et al. successfully printed implants using four different polymers: polylactic acid (PLA), antibacterial PLA (Anti), polyethylene terephthalate glycol (PETG), and poly(methyl methacrylate) (PMMA). The model drug, diclofenac, was poured manually by stopping the process followed by printing of the top layers [45]. The technique was also used for the fabrication of 3D-printed wafers loaded with nanostructured lipid carriers (NLCs) of quercetin and piperine. The wafers were fabricated using FDM (Ultimaker 3, Ulti-maker, Hamburg, Germany) from PVA filament. The NLCs were filled into the wafer manually at room temperature, followed by fixing the wafers to each other to avoid thermal degradation [98]. Linares and colleagues demonstrated the fabrication of printfills (printed scaffold filled with solution) containing ibuprofen as an active ingredient using a combination of FDM and injection volume filling (IVF) techniques. The printfills were manufactured using a REGEMAT 3D V1 printer (Regemat 3D S.L., Granada, Spain) that combined both techniques. The scaffold was made from polylactic acid (PLA) using the FDM, followed by injection of a model drug and delaying the release polymer of dispersions in the PLA structure at room temperature [99] (Figure 11a,b).

Fabrication of capsular-shaped floating devices using FDM and manually inserting tablets of a model drug into the capsule were also reported in the literature. This technique could be used to deliver thermolabile drugs. Charoenying et al. demonstrated fabrication of domperidone-incorporated capsule-shaped floating device. Briefly, the device was produced by using a FDM 3D printer (Prusa i3 MK3, Prusa Research S.R.O., Prague, Czech Republic). The cap was produced from a PVA filament, whereas the body was constructed from a PLA filament. The commercial domperidone tablet was inserted manually into the capsule [100]. Likewise, Cotabarren et al. demonstrated 3DP of PVA capsular devices (CD) for modified drug delivery using a Prusa type (Prusa I3, Hephestos, Buenos Aires, Argentina) FDM printer. The bottom part of the PVA-CD was printed, leaving an open cavity for the filling process. Then, the printing was stopped, and the open shell was manually filled with the active drug, crystalline sodium cromoglycate. Finally, the top part of the PVA-CD was printed, closing the shell and fully sealing the powder [101]. The same technique was reported by Smith et al. [102,103]. Berg et al. and Eleftheriadis et al. used the same technique to fabricate macromolecules at ambient temperature [104,105].

Manually inserting a tablet of a model drug into a gastro-retentive drug delivery system (GRDDS) fabricated with FDM has also been reported. Dumpa and his colleagues developed novel GRRDS using a FDM printer (Prusa I3 3D desktop printer, Prusa Research, Prague, Czech Republic). The shell was prepared from hydroxypropyl cellulose (HPC) and ethyl cellulose (EC)-based filaments, and a directly compressed theophylline tablet was inserted into the core, circumventing the high temperatures involved in the FDM 3D printing process [106]. Acyclovir-loaded gastro-retentive delivery system was also fabricated in the same fashion. The gastro-floating device was printed with commercial PLA filament using a FDM printer (Raise3DN2, Raise3D, Inc., Irvine, CA, USA). The acyclovir tablet was inserted into the floating device manually to allow for sustained release of the drug in the stomach [107]. Similarly, a tablet in device (TID) was fabricated from PVA filament using an FDM printer (CF-12410B, Manli Technology Group Ltd., Hong Kong, China), and a compressed riboflavin tablet was filled into the device manually at room temperature [108]. Manually loading a model drug into a suppository shell produced by FDM was also reported [109,110]. Compartmental dosage forms containing a combination of the anti-tuberculosis drug were also investigated. The shell of the dual-compartmental dosage unit was produced from PLA filament using FDM printing in the vertical position. The printing process was stopped and a known weight of the drug (Isoniazid or rifampicin extrudate, was manually loaded prior to printing the PVA cap. The extrudate of the active drug was prepared by gradually adding API-PEO mixtures to the extruder (DSM, ^®^XPLORE, Geleen, The Netherlands) and extruded at 80 °C [111]. Likewise, Matijasic et al. developed a drug-loaded multicompartmental PVA capsule for drug delivery. The capsules were printed using the Flash forge Inventor I printer equipped with a 0.4 mm nozzle from PVA filament. Model drugs, dronedarone hydrochloride, and ascorbic acid in powdery form were filled into the printed capsules [112].

Tiboni et al. reported automatic filling of drug delivery devices produced using an Ultimaker 3 printer (Ultimaker, Geldermalsen, The Netherlands). The device was loaded with a model drug, cannabidiol, containing a formulation prepared using two different model nanocarriers (i.e., polymeric nanoparticles and liposomes). The formulations were loaded onto the devices through two syringes mounted on two syringe pumps (Aladdin syringe pump, WPI Europe, Friedberg (Hessen), Germany) through polyethylene tubing. The organic solvent used in the formulation was evaporated under a stream of nitrogen [113]. Similarly, Okwuosa and co-workers reported a fully automated additive manufacturing process that combined FDM and liquid dispensing to fabricate individualised dosage forms on demand (Figure 12). They modified a dual FDM 3D printer to include a syringe-based liquid dispenser. Polymethacrylate shells (Eudragit EPO and RL) for immediate and extended-release were fabricated using FDM 3DP and simultaneously filled using a computer-controlled liquid dispenser loaded with model drug solution (theophylline) or suspension (dipyridamole). Dipyridamole suspension was prepared at room temperature while theophylline solution was prepared by mixing components at 65 °C [114].

Beck and colleagues coupled two important technologies, nanotechnology and 3DP, to produce 3D-printed tablets loaded with polymeric nanocapsules. In short, the drug delivery device (tablet) was prepared by FDM from poly(e-caprolactone) (PCL) and Eudragit1RL100(ERL) filaments with or without a channelling agent(mannitol). The prepared tablets were soaked in a deflazacort-loaded nano-suspension for 24 h at room temperature. Finally, the tablets were removed and dried at 30 °C for 24 h [115].

Techniques that coupled FDM with melt casting were also reported in the literature and could be effectively used to fabricate dosage forms containing thermosensitive drugs. A polypill containing thermosensitive drugs, aspirin, and simvastatin was prepared by integration of melt cast method and FDM. The drugs were mixed with molten PEG 6000, glycerin, and silica using a magnetic stirrer at 1000 rpm at 55 °C. Finally, the molten mixture containing the model drugs was directly injected into a compartment of the 3D-printed scaffold fabricated from Eudragit L100-55 filament at an extrusion temperature of 178 °C using an insulin syringe. PEG 6000 was used as the main polymer due to its low melting point temperature (70 °C), which helped the drug to be preserved from overheating and degradation [44]. Ajmal and co-workers also illustrated the use of FDM 3D-printed moulds to cast tablets forming different shaped tablets. The casting formulation was prepared by mixing the model drug, indomethacin, with PEG 600 croscarmellose sodium (SCC) and hydrated HPMC at room temperature. Finally, the casting formulation was poured into the 3D-printed moulds. Excess casting formulation was removed using a spatula to provide the casted tablet with a smooth surface. The casted tablets were left to dry at room temperature for 24 h. Once dried, the printed cylinder was removed from each mould, and the tablets were detached using a scalpel [116].

##### FDM and Inkjet Printing

Eleftheriadis et al. reported a combination of IP and FDM to avoid thermal degradation associated with FDM. The polymeric platform was printed using an FDM printer (MakerBot Inc., Brooklyn, NY, USA) using a filament obtained from HME at a temperature of 172 °C. The ink containing a mixture of ibuprofen, ethanol, and propylene glycol was loaded onto a film produced by FDM at room temperature [117]. In another work, Eleftheriadis and co-workers used the same principle to fabricate mucoadhesive buccal films for local administration of ketoprofen and lidocaine hydrochloride. The HPMC-based film was fabricated using a Makerbot Replicator 2X 3D printer (Mak-erBot Inc., Brooklyn, NY, USA) from the ketoprofen-loaded filament. An array of ink containing different concentrations of lidocaine was deposited onto the fabricated film in 10 repetitions at room temperature highlighting IP coupled with FDM could be used as a good strategy for fabricating thermolabile drugs [118]. An innovative drop-on-powder (DoP), an inkjet technique, was used to produce oral tablets loaded with an anticancer model drug, 5-fluorouracil (FLU). All tablets were printed using a commercial FDM printer (ZCorp 3D printer) with modification. The resultant tablets were subject to coating with various polymeric solutions containing the drug [119].

##### FDM and Supercritical Fluid Technology (SFT)

Coupling FDM and SFT could also be used to incorporate thermosensitive drugs as it does not involve heat elements. Schmid et al. coupled FDM with SFT to develop 3D-Printed Controlled Drug Release Dosage Forms. Scaffolds with varying pore sizes were made from polylactic acid (PLA) using a FDM 3D printer (German RepRap GmbH, Feldkirchen, Germany). The 3D-printed drug carriers were then loaded with ibuprofen as a model drug, employing the controlled particle deposition (CPD) process from supercritical CO_2_ at room temperature, highlighting the benefit of the technology to incorporate thermosensitive drugs [120].

##### SLA and Inkjet Printing

SLA and inkjet were also coupled to exploit the strengths of both methods. For instance, the high resolution of SLA was useful in developing microneedles for the delivery of insulin transdermally [121]. However, the curing process involved exposure to temperatures ranging from 40 to 60 °C. Consequently, insulin was loaded onto the needles by inkjet printers at room temperature, avoiding any risk of thermal degradation and ensuring high accuracy and drug uniformity offered by the digital quality of 3DP [122]. However, 3D-printed microneedles loaded with rifampicin using SLA only were attempted at a temperature of 29.8 °C [123]. In another study, both techniques were also merged to fabricate precisely positioned drug depots within a polymer matrix to provide controlled release profiles using bovine serum albumin as a model drug [124].
pharmaceutics-13-01524-t001_Table 1Table 1Summary of polymers and printer types used for printing thermosensitive drugs.Printing TechniquePrinter NameDrugPolymerReferenceSSEBuilt in houseVancomycin ceftazidime PCLNano-hydroxyapatite[66]SSEVelleman K8200 (Velleman Inc., Fort Worth, TX, USA)TheophyllineHPMC[70]SSEFochif Mechatronics Technology Co., Ltd. (Shanghai, China)PuerarinHPC[62]SSEMAM II, Fochif Mechatronics Technology Co., Ltd. (Shanghai, China)LevetiracetamHPC[71]SSE3D-Bioplotter, EnvisionTec (Gladbeck, Germany)LevetiracetamPolyvinyl acetate/polyvinylpyrrolidone HPMC [68]SSEFochif Mechatronics Technology Co., Ltd. (Shanghai, China)Ginkgolide HPMC Methocel K4MHPMC Methocel E5LVMicrocrystalline cellulosePVP[63]SSEInkredible, Cellink (Gothenburg, Sweden)CaffeineHPMCPullulan[72]SSEBiobot 1, Allevi (Philadelphia, PA, USA)DapagliflozinCapryol 90Poloxamer 188PEG 6000, PEG 4000, PEG 400Cremophore EL[69]SSEMAMII; Fochif Mechatronics Technology Company, Ltd. (Shanghai, China)LevetiracetamPVPCarboxymethylcellulose sodium[59]SSECellink (Gothenburg, Sweden)FenofibrateKolliphor ELCaptex 355 EP/NFCapmul MCM EP[57]SSEHyrel System 30 M, Hyrel 3D (Atlanta, GA, USA)TheophyllineEudragit^®^ RL100 and RS100[58]SSEROKIT INVIVO (Seoul, Korea)RiboflavineCrosslinked tyramine–modified methylcellulose[73]SSEMAMII, Fochif Mechatronics Technology Co., Ltd. (Shanghai, China)DipyridamoleHPMCMicrocrystalline cellulose[60]SSE3-Donor^®^, Life SI (Córdoba, Argentina)RicobendazoleGelucire[61]SSEHyrel System 30M, Hyrel 3D, (Norcross, GA, USA)TheophyllinePVA[56]SSEK8200 Velleman (Gavere, Belgium)-PLGA[64]FDMNSDoxycyclinePCL[76]FDMUltimaker 3 Extended, The NetherlandsIsoniazid, rifampicin BPolyethyleneoxide[111]FDMMultirap M420 (Illmensee, Germany)QuinineEudragit^®^RSPCLPoly(l-lactide) (PLLA)Ethyl cellulose (EC))[77]FDMMakerBot^®^ Replicator 2 desktop 3D printer (Brooklyn, NY, USA)HaloperidolGlutaric acidPolyvinylpyrrolidone-vinyl acetate copolymer (Kollidon^®^ VA64) Hydroxypropyl methylcellulose HME 15cP [81]FDMMakerBot Replicator 2X Desktop, MakerBot Inc. (Brooklyn, NY, USA)RamiprilKollidon VA64Kollidon 12PF[78]FDMGeo technology (Incheon, Korea)RifampicinPCL[65]FDMMakerbot Industries, Brooklyn, NY, USATheophyllinePVP[74]DLPWanhao Duplicator 8 (Zhejiang, China) AtomoxetinePEGDAPEG 400[90]DLPGizmo^®^ 3D printer, Gizimate^®^ 130, Queensland, Australia)FluticasonePoly(caprolactone-dimethacrylate)[94]DLPPico 2 HD (Asiga, Sydney, Australia),Acetyl-hexapeptide 3 Polyethylene glycol diacrylate (PEGDA) and vinyl pyrrolidone (VP)[96]DLPKudo 3D printer (Dublin, CA, USA)Doxycycline, vancomycin cefazolinPEGDA[86]DLPAsiga MAX X27, Asiga Ltd. (Alexandria, Australia)Diclofenac sodiumPEGDA[97]DLPDuplicator 7, (Wanhao, Zhejiang, China)Atomoxetine hydrochloridePEGDA/PEG 400[91]DLPAnycubic Photon 3D (Shenzhen, China)Ascorbic acidPEGDMA[83]DLPForm 1 + SLA 3D printer (Formlabs Inc., Somerville, MA, USA)Irbesartan, atenolol, hydrochlorothiazide, and amlodipinePEG 300/diphenyl(2,4,6-trimethyl-benzoyl) phosphine oxide/PEGDA[85]DLPDLP^®^ Discovery™ 4100, Texas Instruments, Austin, TX, USA TheophyllinePEGDA[88]Drop-on-powderZCorp printer (Z-Corporation, Rock Hill, SC, USA)5-FluorouracilSoluplus[119]SLAForm 1+ (Victoria, Australia)Paracetamol, naproxen, caffeine, aspirin, prednisolone, chloramphenicolPEGDA[25]SLADuplicator 7 (Wanhao Zhejiang, China)IbuprofenPEGDA[93]SLADupicator 7 (Wanhao, Zhejiang, China)ParacetamolPEGDA[92]SLAFormlabs Form 2 SLA (Formlabs Inc., Somerville, MA, USA)ParacetamolAspirinPEGDAPCL Triol[18]SLAFormlabs 1 + SLA (Formlabs Inc., Somerville, MA USA)ParacetamolPEGDA[33]SLA Formlabs 1 + SLA (Formlabs Inc., Somerville, MA USA)IbuprofenPEGDA[84]FDM/SSEUltimaker3, The NetherlandsPropranololPLA/sodium alginate[67]FDM/BJCanon MG2950 thermal inkjet printer (Canon Inc., Athens, Greece)IbuprofenPropylene glycol[117]FDM/Ink jetCanon MG2950 (Canon Inc. Athens, Greece)LidocaineketoprofenPropylene glycol (PG)[118]SLA/Ink jetBuilt in houseBovine albumin serumPEGDA[124]

### 3.3. Types of Dosage Forms

Regarding the types of dosage forms produced, tablets were the most common manufactured formulations (52%, *n* = 42). In addition, drug-eluting constructs such as implants and scaffolds, among others, were reported in 24.6% (*n* = 20) of the studies. Other dosage forms such as capsules, oral films, and microneedles were also among the produced dosage forms, highlighting the application of the methods across different ranges of dosage forms (Figure 13).

### 3.4. Drying Temperature/Solidification Process

Drying/solidification is an important step in producing dosage forms with good mechanical characteristics. This is particularly true for techniques that involve the use of heat or solvent to extrude the formulation. If the material is extruded using heat, this can be done immediately prior to extrusion, and the solidification process will be determined by the cooling of the material. It requires optimising printing temperature to obtain an adequate material viscosity and solidified dosage forms. The use of different solvents during the formulation of semi-solid feed requires solvent evaporation to ensure the solidification of dosage forms. Hardening of the dosage forms can also be achieved by photopolymerisation where monomers/oligomers cross-link upon exposure to a light source (e.g., UV light) in the presence of a photoinitiator. Drying represents an important step in the manufacture of thermosensitive drugs as the process could lead to degradation. Since FDM uses thermoplastic polymers that usually solidify at room temperature or temperature below the printing temperature, the drying step is not essential.

Different drying methods and temperatures were reported for thermosensitive drugs produced by SSE (Table 2). Few studies dried the final product at room temperature to remove the solvents. For instance, Mengsuo et al. dried the 3D-printed LEV tablet at room temperature for 48 h [59]. Similarly, theophylline tablets produced by solvent-free extrusion techniques were solidified at room temperature [58] (Figure 14).

Few other studies used heat dryers to dry the prepared dosage forms. Theophylline tablet prepared by temperature and solvent facilitated extrusion was dried for 2 h at 50 °C using a Binder Drying chamber 9010 (Binder GmbH, Germany) [56]. Jenny and her colleagues dried fenofibrate tablets in a vacuum oven (Vacutherm, Thermo Fisher Scientific, Waltham, MA, USA) at room temperature overnight [57]. The 3D-printed dipyridamole tablet was dried in an oven at 40 °C for 12 h. In another study, a small amount of MEK solvent detected in the 3D-printed scaffold was dried by placing a vacuum flask attached to a vacuum line and leaving it to dry for one week to obtain a dry scaffold with good mechanical properties [64,65] (Figure 15).
pharmaceutics-13-01524-t002_Table 2Table 2Drying conditions of some 3D-printed drug dosage forms produced using the SSE technique.Year of PublicationDrug and Dosage FormDrying TechniqueDrying TemperatureDrying TimeReference2018Levetiracetam tabletOpen-airRoom temperature48 h[60]2020Theophylline tabletDrying chamber50 °C2 h[56]2021Fenofibrate tabletOpen-airRoom temperature24 h[57]2020Drug-eluting constructVacuum flask attached to a vacuum lineRoom temperatureOne week[125]2020Ricobendazole tabletPrinting platform25 °CNS *[61]2019Catechin tabletAir driedRoom temperatureNS *[126]Freeze-dryer

2018Dipyridamole tabletOven40 °C12 h[60]2021Caffeine tabletDesiccatorNS48 h[72]2021Vancomycin and ceftazidime tabletOvenRoom temperature72 h[66]2020Levetiracetam tabletDry airRoom temperature48 h[71]2019Levetiracetam tabletPrinting bed27 °CNS *[51] 2019Puerarin tabletOven40 °C12 h[62]2020Levetiracetam tabletPrinting bed27 °CNS *[68]2021Dapagliflozin tabletPrinting bedRoom temperatureNS *[69]2020ScaffoldGlass plate37 °CNS *[73]2019Ginkgolide tabletOvenNS *12 h[63]2021Propranolol tabletAirRoom temperatureNS *[67]Oven40 °C7 hMicrowave200 W10 minVacum/desiccator
6 h/overnight* Not specified.

## 4. Conclusions and Future Perspectives

3DP is a revolutionary technique that provides the possibility of manufacturing patient-personalised formulations, offering several benefits such as improved safety, decreased cost, and enhanced adherence to treatments. The FDA approval of the first 3D-printed tablet (Spritam^®^) in 2015 significantly increased interest in the technology to produce patient-centric dosage forms. Despite exciting advantages, most 3D printing technologies use higher temperature/source of light that could degrade thermolabile drugs.

SSE is found to be the most widely (24%) used technique for the production of thermolabile drugs/scaffolds. SSE offers the advantage of printing dosage forms at a lower temperature. The use of the FDM printing technique, known for its versatility, for fabricating thermosensitive drugs has been limited by its use of higher temperatures. Several attempts were made to fit FDM to produce thermolabile drugs, among which proper selection of polymer and excipients with low glass transition temperatures as well as combining with other printing techniques such filling and inkjet printing are the major examples. Production of the delivery system using the FDM technique and manually/automatically filling the delivery system with the active drug could be used for the preparation of thermosensitive drugs. DLP, SLA, and IJ printing techniques either alone or in combination with other techniques could also be used for the fabrication of thermosensitive drugs. It is worth noting that degradation of some drugs was reported due to the light source used in the process. Degradation of the drugs was also avoided by drying the drugs at lower or room temperature. In order to understand the importance of thermosensitive drugs for the treatment of deadly diseases and their cost, as well as the promising future of 3D printing, more research into the design of FDM printers that circumvents the thermal degradation is required.

Despite the rapidly building momentum and promising future of 3D printing in the pharmaceutical field, technical and regulatory challenges that hinder its implementation need to be considered. For instance, enhancing reproducibility and appearance of the finished products requires further improvement. More research into suitable polymers, surface finishes, and the design of better printers is required to improve the quality of the finished products. The regulatory landscape is another huge challenge hindering the implementation of the technology. A timely and clear regulatory pathway for 3D-printed drug dosage forms is warranted to benefit from the advantages it brings to personalised medicine.

Considering resources needed to implement the technology at the point of care is also essential. For instance, the cost of printers, experts in 3D printing, and digital design required at each point of care requires paramount attention. Few studies have suggested artificial intelligence [127], design of experiment set-up [128], and algorithms as practical solutions for the design of pharmaceutical products, thereby reducing the workforce required at each care point. Further studies that address the implementation and scalability of the technology are warranted.

Last but not least, whilst there is a consensus about the manufacturing and environmental benefits of additive manufacturing compared to conventional methods, some studies have raised concerns about the impact of 3DP on the environment, such as energy consumption [129] and volatile organic solvent emission upon filament fabrication [130]. Nonetheless, standard test methods that determine and analyse the effect of each process on the environment are required.

## Figures and Tables

**Figure 1 pharmaceutics-13-01524-f001:**
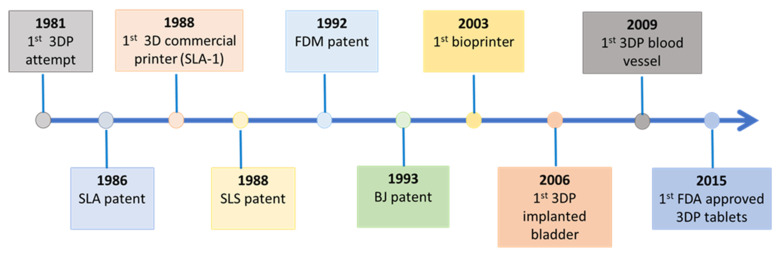
Invention and development of different 3DP techniques and products.

**Figure 2 pharmaceutics-13-01524-f002:**
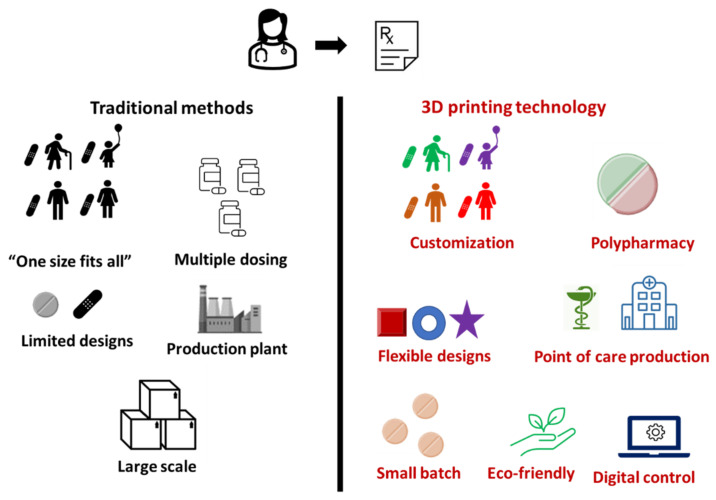
Comparative advantages of 3DP over traditional methods.

**Figure 3 pharmaceutics-13-01524-f003:**
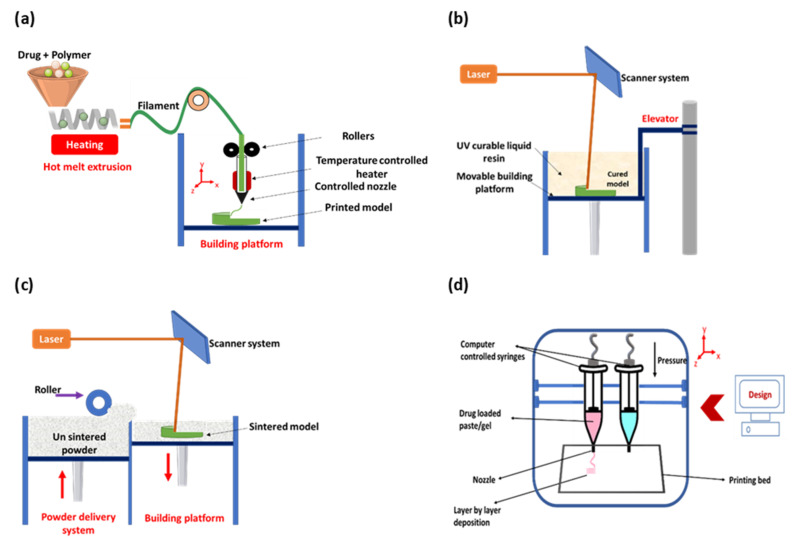
A schematic diagram (**a**) showing the different parts of the FDM printer: filament spool, heated printer nozzle, and printing platform; (**b**) showing the different parts of the SLA printer: laser/UV source, resin tray, and printing platform; (**c**) showing the different parts of the SLS printer: powder roller, laser beam source, laser scanner, and fabrication piston; and (**d**) showing the different parts of the SSE printer: syringes, nozzle, and printing bed.

**Figure 4 pharmaceutics-13-01524-f004:**
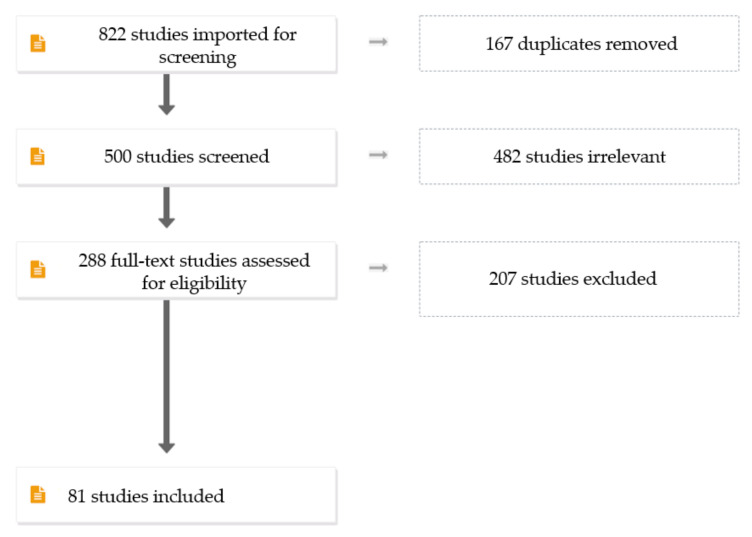
Flowchart showing publications selection process.

**Figure 5 pharmaceutics-13-01524-f005:**
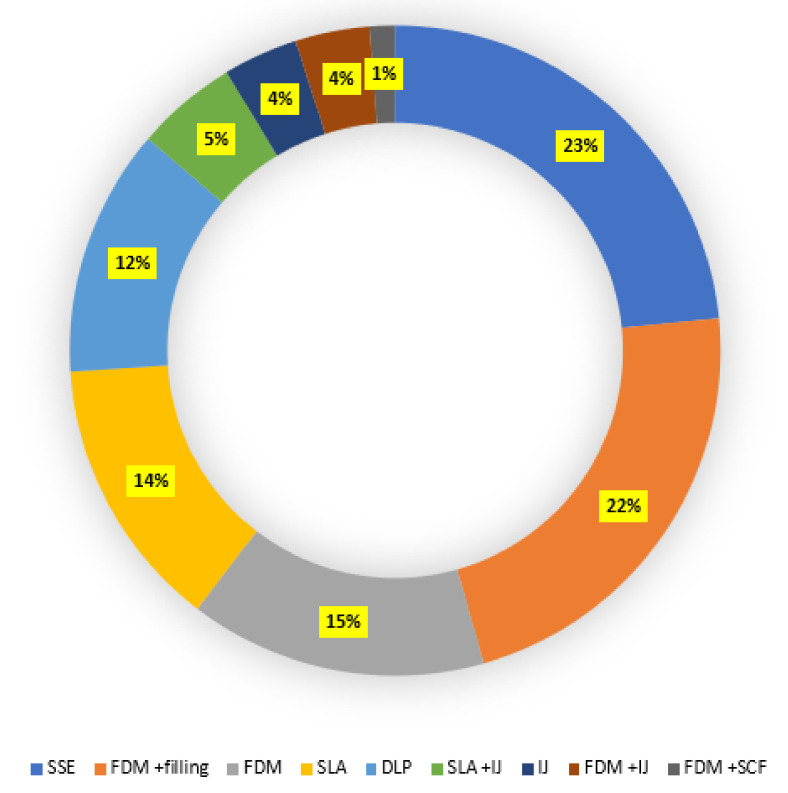
3DP techniques used in the articles included in the systematic review (IJ, inkjet; SCF, supercritical fluid technology).

**Figure 6 pharmaceutics-13-01524-f006:**
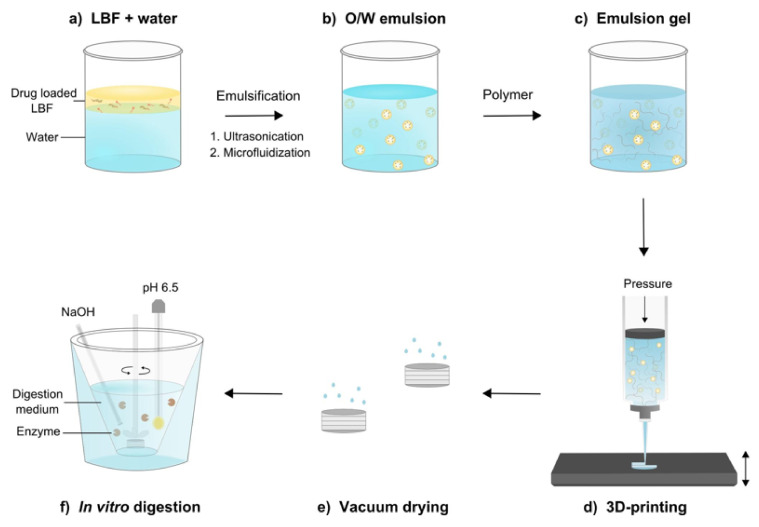
Schematics illustrating the preparation of printable emulsion gels, 3D printing by semi-solid extrusion (SSE), and in vitro digestion of 3D-printed tablets. (**a**) Drug-loaded lipid-based formulation (LBF) was added to water, followed by (**b**) a two-step emulsification process. (**c**) Polymers were added to emulsified LBFs to generate printable emulsion gels. (**d**) The emulsion gels were 3D-printed by SSE into tablets, and (**e**) vacuum-dried. (**f**) 3D-printed tablets were digested in an in vitro lipolysis set-up to quantify the release of free fatty acids (FAs). Figures reproduced with permission from [57].

**Figure 7 pharmaceutics-13-01524-f007:**
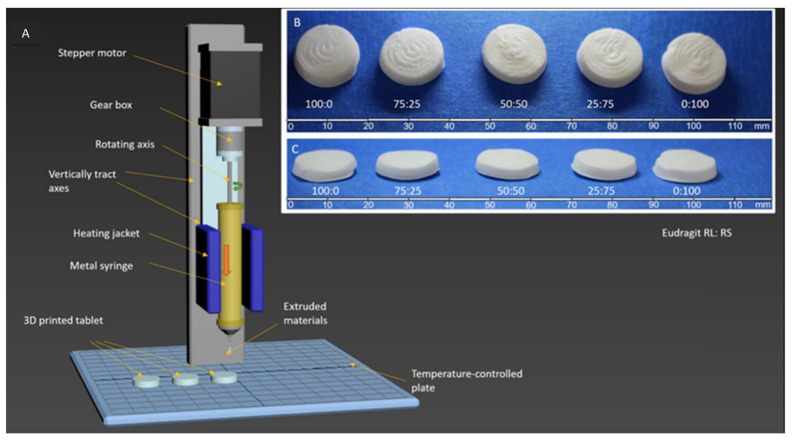
(**A**) Set-up for direct extrusion 3D printing. The printer is equipped with a metal syringe surrounded by a temperature-controlled heating jacket. The syringe is fitted with a luer-lock stainless steel needle (G18), and the pharmaceutical ink (compressed powder) is added. The ink is then extruded by a piston pushed by a computer-controlled stepper motor equipped with gear to produce 3D-printed tablet. (**B**) Top and (**C**) side photographs of 3D-printed tablets based on Eudragit RL: RS: 100:0, 75:25, 50:50, 25:75, and 0:100. Figures reproduced with permission from [58]. Copyright Elsevier, 2021.

**Figure 8 pharmaceutics-13-01524-f008:**
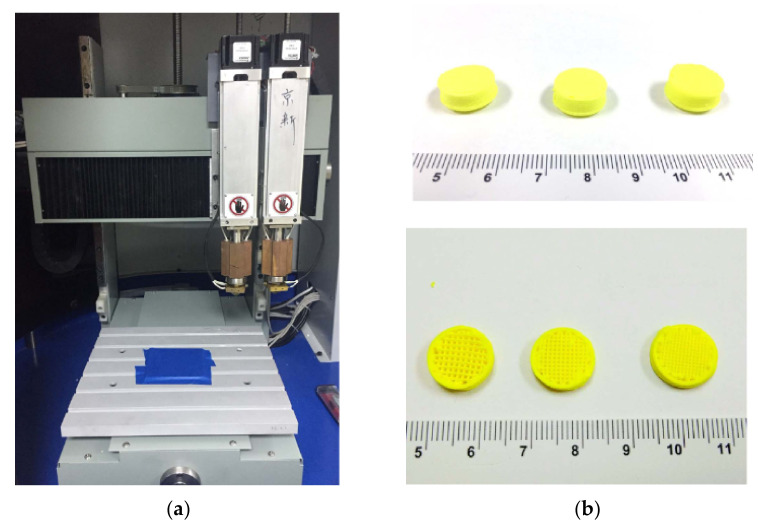
(**a**) Commercial extrusion-based dual-nozzle 3D printer. (**b**) The 3D-printed gastro-floating tablets with different infilling percentages and the section of the tablets; the infilling percentages are 30%, 50%, and 70% from left to right. Images reproduced with permission from [60]. Copyright Elsevier, 2018.

**Figure 9 pharmaceutics-13-01524-f009:**
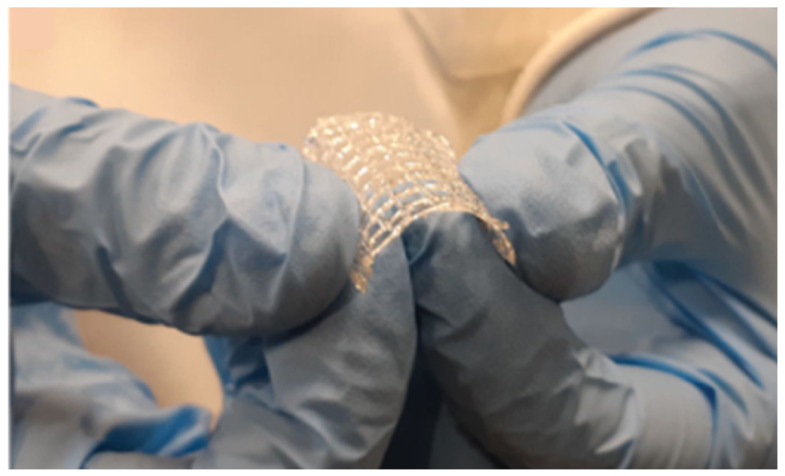
3D-printed rifampicin-loaded scaffold showing great mechanical flexibility and integrity upon bending. Images reproduced with permission from [65].

**Figure 10 pharmaceutics-13-01524-f010:**
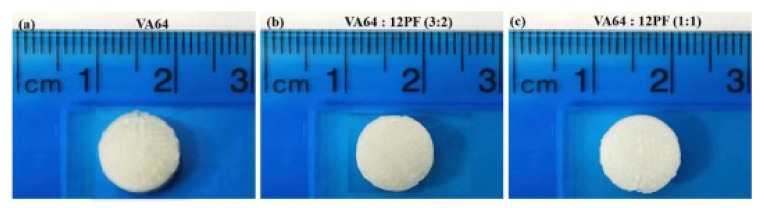
Pictures of ramipril printlets. (**a**) VA64, (**b**) VA64:12PF (3:2) and (**c**) VA64:PF12 (1:1). Image reproduced with permission from [78]. Copyright Elsevier, 2018.

**Figure 11 pharmaceutics-13-01524-f011:**
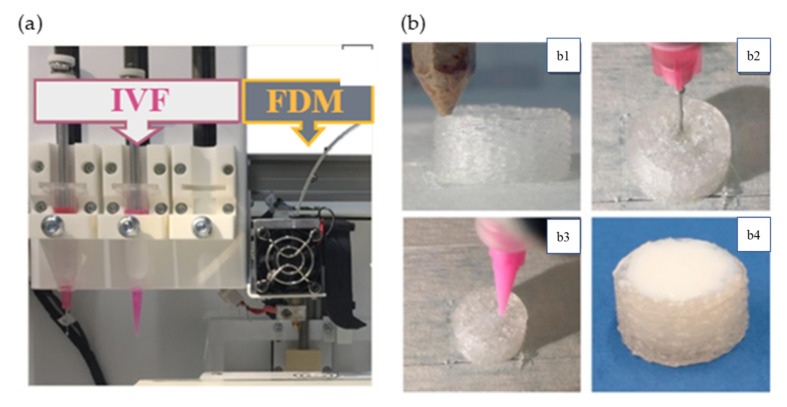
(**a**) Integration of FDM with IVF and (**b**) (**b1**) extruder of FDM technology. (**b2**) Syringe of IVF technology injecting the drug-loaded gel. (**b3**) Syringe of IVF technology injecting the delaying release polymer. (**b4**) Obtained images of final printfill of the tablets. Images reproduced with per-mission from [99]. Copyright Elsevier, 2019.

**Figure 12 pharmaceutics-13-01524-f012:**
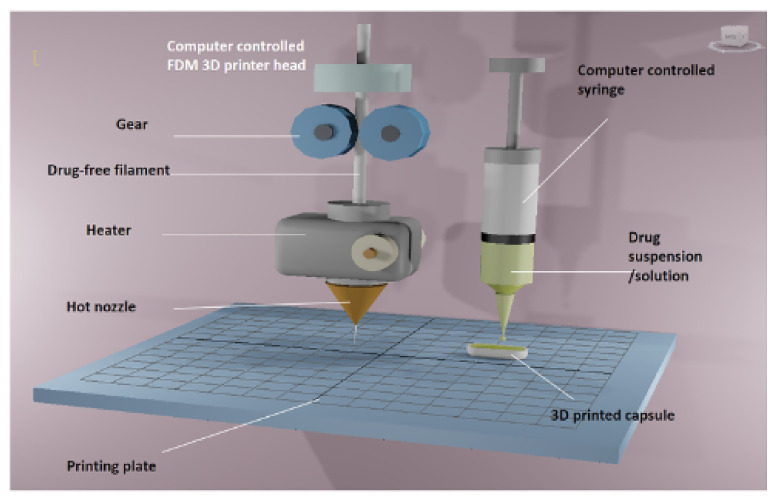
Schematic illustration of the fabrication of 3D-printed liquid capsule. A dual-head 3D printer was modified by replacing the right-hand nozzle with a syringe dispenser. The FDM nozzle head was loaded with HME processed API-free filament of immediate or extended-release properties whilst drug solution or suspension were dispensed using syringes of variable sizes and nozzle diameters. Images reproduced with permission from [114]. Copyright Elsevier, 2018.

**Figure 13 pharmaceutics-13-01524-f013:**
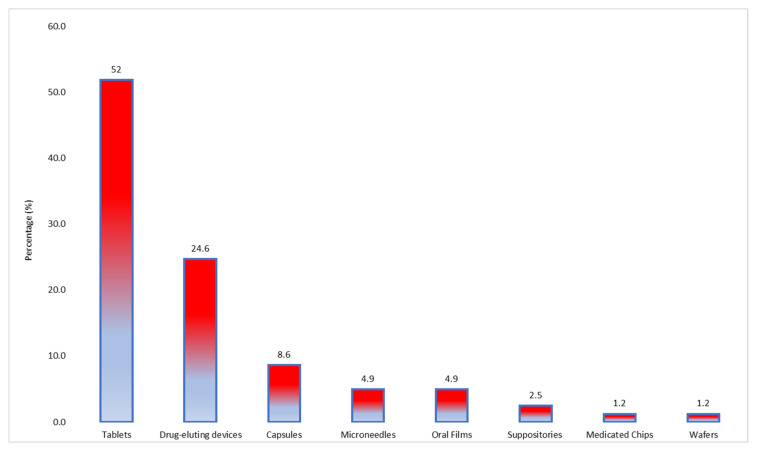
Dosage forms reported in the studies included in the systematic review.

**Figure 14 pharmaceutics-13-01524-f014:**
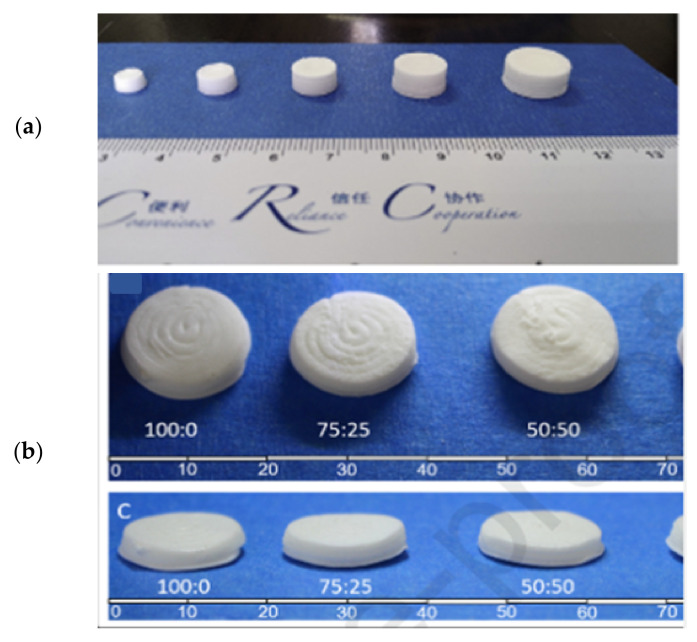
Dried 3D-printed levetiracetam tablets (**a**) and theophylline tablets (**b**) solidified at room temperature. Images reproduced with permission from [58,59]. Copyright Elsevier, 2021 & 2019.

**Figure 15 pharmaceutics-13-01524-f015:**
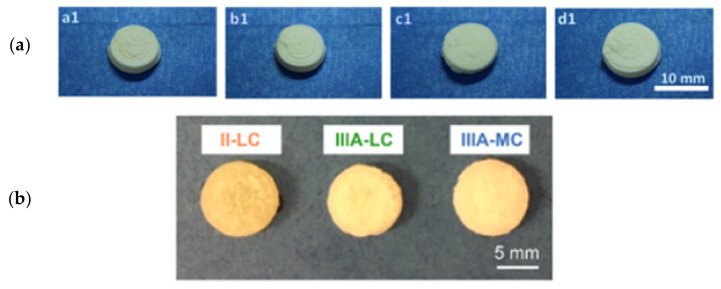
3D-printed tablets of (**a**) theophylline (dried at 50 °C) ((**a1**) PVA and sorbitol, (**b1**) PVA and lactose and (**c1**) PVA and D-mannitol and (**d1**) PVP and lactose) and (**b**) scaffold (dried using vacuum flask at room temperature). Images reproduced with permission from [56,57]. Copyright Elsevier, 2020.

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
