# Peer review of "3D Printing of Thermo-Sensitive Drugs"

_pharmaceutics, 2021, doi:10.3390/pharmaceutics13091524_

Round 1

Reviewer 1 Report

The manuscript submitted by Abdella et al. is a review article dealing with the potential of 3D printing technologies in formulating thermosensitive actives. This is certainly an interesting topic, with limited discussion in the available literature. The review provides crucial information that is required to acknowledge the concept and current challenges of pharmaceutical 3D printing with thermolabile drugs. However, more details and critical analysis is required in some parts of the manuscript to improve its content. The following comments should be addressed before I would support publication:

Comments

  1. Lines 3-12: Revision is required on the symbolic representation of the roles of the authors. Some symbols are currently missing in the authors list, e.g., the corresponding author.

  1. Line 67: Please revise the phrase to “according to”.

  1. Lines 76-80: The authors introduce “Figure 3”, while Figure 2 is not mentioned in the manuscript, yet. In order to keep the current flow of Figures in the text, I recommend that the authors provide some references here.

  1. Lines 75-135: The authors are advised to discuss the difficulties of implementing such scenarios, with distributed 3D printers for point of care production or for personalized dosing. There should also exist distributed specialized workforce, e.g., 3D printing and digital design experts, at each point of manufacturing, in order to achieve this concept in hospitals and other decentralized manufacturing hubs. I recommend that the authors acknowledge previous published works that discuss the use of algorithms in the pharmaceutical applications of 3D printing, e.g., for automated digital design and 3D printing of individualized dosage forms, or with the aid of artificial intelligence. Alternatively, this can be discussed in the Conclusions and Future Perspectives section at the end of the manuscript.

  1. Line 135: I recommend that the authors further mention other ecological issues that exist for 3D printers, e.g., the emissions of volatiles from FDM printers.

  1. Line 146: Please revise the name of the marketed 3D-printed product “Spritam”.

  1. Lines 158-161: Please revise this sentence for grammar and syntax. It currently makes no sense.

  1. Lines 200-208: Please mention that a rise in temperature is commonly utilized to achieve the required viscosity of the feedstock in SSE.

  1. Lines 230-236: Why only the term of semi-solid extrusion printing was added in the keywords? Using additional keywords, e.g., FDM, could also return more results (please also see on my next comments some works that are missing from the current review).

  1. The term “pressure-assisted microsyringe” is an equal alternative to SSE that is commonly used in the pharmaceutical field, whereas in most of the published works this technology has been used due to the absence of high temperature. Please also search for works that use this term.

  1. Lines 389-390: Autodesk Inc is a company that produces software for digital design and not 3D printers. Please check the cited work and mention the correct 3D printer.

  1. I recommend that the authors provide a separate section for published works that combined FDM with other technologies. The reason is that, in many cases, the manufactured drug delivery systems are able to host thermosensitive drugs. Please see further recommendations, below.

  1. Regarding the use of FDM printing, the authors are advised to discuss and cite previously published works on the use of compartmental and multicompartmental drug delivery systems. In most of the cases, the various compartments were manually or automatically filled with the drug, e.g., https://doi.org/10.1016/j.jconrel.2017.10.003 and https://doi.org/10.1016/j.ijpharm.2019.118494, thus rendering these approaches suitable for thermolabile drugs. Moreover, a similar system has been mentioned in the literature for oral peptide delivery (https://doi.org/10.1016/j.xphs.2020.10.066).

  1. Another aspect is the coupling of FDM with electrospinning or Supercritical fluid technology. In these cases, also, the concept can be used for thermolabile drugs.

  1. The combination of FDM and inkjet printing has been further applied for dual loading of formulations with drugs that present different thermal degradation profiles (https://doi.org/10.1016/j.xphs.2020.05.022).

  1. There are two published articles in the current Special Issue that describe the low-temperature thermal processing (https://doi.org/10.3390/pharmaceutics13060907) or the use of porous materials that provided increased thermal stability of the drug (https://doi.org/10.3390/pharmaceutics13071096). The authors are advised to acknowledge these works.

  1. References 58 and 59 are duplicates.

  1. I recommend that the authors expand the section of Conclusions and Future Perspectives. This section fails to present an authors’ opinion or perspective on the potential of 3D printing for thermolabile drugs.

Reviewer 2 Report

The review manuscript “3D printing of thermo-sensitive drugs” by Abdella, et al. reviews and summarized 3D printing technology evolved in pharmaceutical industry and looked into thermosensitive drugs in the market with a strict temperature control processing and presents challenges in 3D printing procedures and its future directions for thermo-sensitive formulations.

The manuscript structure started as a review paper however towards end of page 6, it changes its format to a research paper (to describe how the literature search output was used). If the authors are trying to share their findings as a literature research article, it cannot be a review article.

For instance, what is the question of the reader from reviewing this article? Would it be how they collected presented data or getting a comprehensive understanding of the current state art for 3D printing technology of thermo-sensitive drugs in pharmaceutical industry?

I found several formatting issues with prepared figures:

  • Figure 2. Missing scale bar for provided images. This information could have been extracted from Figure 1 from Tagami, T., Ito, E., Kida, R., Hirose, K., Noda, T., & Ozeki, T. (2020). 3D printing of gummy drug formulations composed of gelatin and an HPMC-based hydrogel for pediatric use. International Journal of Pharmaceutics, 120118. doi:10.1016/j.ijpharm.2020.120118
  • It would be beneficial to the reader to provide more details for figure legends so that the reader can fully understand the content of your figure without having to refer to the main text (example Figure 4, 7, 8)
  • General formatting: For instance, in Figure 6. 3DP techniques used in the articles included in the systematic review, not all numbers have percentage sign.
  • There is a point at the end of phrase “et al.” which are missing throughout the manuscript for example Mengsuo et al (Line 441), Pan et al (Line 111), Emad et al (Line 340)

From review structure layout, there are several topics authors did not include in their review paper that can be very beneficial to the readers including the effects of 3D printing parameters on the drug release kinetics, summary of monomers and oligomers used for incorporation of thermo-sensitive drugs, shelf-life comparison (between existing manufacturing of drug formulations vs. 3D printed formulations of same kind), scalability (current manufacturing vs 3DP), and finally list of current 3D printing platforms companies which designed/customized printers for this industry.

Reviewer 3 Report

This paper evaluated a very interesting topic, such as suitability for 3d printing of thermosensitive drugs. Since most of the 3d printing techniques imply processing at elevated temperatures, preparation of formulations with thermosensitive drugs remains challenging. In general, the manuscript is well written, but the main drawback is in the methodology of this review. Authors limited literature search only on the references which necessarily includes terms thermosensitive, thermolabile, etc., which excludes some relevant papers describing 3d printing techniques which are also suitable for thermosensitive drugs, although this is not explicitly stated in these studies. For example, stereolithography, digital light processing, and binder-jet 3d printing do not include processing at high temperatures which makes these techniques very suitable for thermosensitive drugs. Therefore authors should include additional references which describe 3d printing techniques suitable for thermosensitive drugs. Additionally, there are some language and typographic mistakes, missing spaces, or extra spaces in the text, which authors should correct in the revision. Keywords should be more informative.

Reviewer 4 Report

The topic covered by this review is interesting and will attract the interest of the readers, as additive manufacturing is continuously developing and probably will be more explored for drugs production. The authors must check the manuscript for typo. Please correct the capitalized letters within sentences (ex. L 322 “Firstly, Hydroxypropyl methylcellulose”). With few exceptions, the text is well written and can be considered for publication after major revision.

Introduction: L138-L148 My suggestion is to move this sentence in the first part of the introduction. It doesn’t fit the sub-title “Waste minimization”.  

L80: Please correct “Figure 3” to “Figure 2”. I suggest to move this figure (“Comparative advantages of 3DP over traditional methods”) before Figure 2, according to the flow of the discussion

It seems that all the sub-titles of the section “1.1. Advantages of Three-dimensional Printing (3DP) in dosage form production” has the same no. – “1.1.1”.

L111, L112 – this sentence came from nowhere; is not linked with the previous one (which, in my opinion, is not so relevant, as the precise dosing of the drug is easily achieved with the technology used currently for the mass-production of the drugs)

L119-L120: what do you mean ? Figure 2 : what is the connection between the shape and drug release ? more information should be provided in text

Fig 8 – It seems the authors cropped too much the original figures. Some words are incomplete.

L320: please correct “water-solublility”

L454 – MEK was previously abbreviated (L342)

L365: is not Poloxamer ?

L385: please add abbreviation (PLA), as the polymer is mentioned for the first time and used latter

Is there any report concerning drug release profile on 3D printed tablets ?

Round 2

Reviewer 1 Report

The manuscript submitted by Abdella et al. was properly revised, considering this reviewer’s comments. However, there is now a new issue that must be addressed before I would support publication.

  1. The authors must carefully check the revised version for complete and accurate compliance of the References list with the references mentioned in the manuscript. After a quick search of the references mentioned in the text with “et al.”, I found that a proper check of the cited works one-by-one must be performed all over the manuscript. For example, reference “10” at Line 81 does not refer to “Sachs et al”; reference “98” at Line 570 does not refer to either “Staffan et al” or “Eleftheriadis et al”. I assume that there is total mismatch all over the manuscript between the references in the text and the list, due to the revisions that have been made. Please revise all references one-by-one, accordingly.

Reviewer 2 Report

I found the revised manuscript "3D printing of thermo-sensitive drugs" by  S. Abdella , et al. sufficiently addressed all raised concerns and properly amended the text to reflect those suggestions.    I only found one minor formatting issue in title as it started with "Title: 3D printing of thermo-sensitive drugs" and the word title needs to be omitted.   I recommend the editorial team to accept the manuscript in current version . 

Reviewer 4 Report

The manuscript has been consistently improved. I congratulate authors for their work.

Author Response

Thank you very much. 

Round 3

Reviewer 1 Report

The authors have successfully addressed this reviewer's comments. 

Reviewer 2 Report

I found the revised review manuscript "3D printing of thermo-sensitive drugs" by  S. Abdella , et al. adequately addressing all raised concerns and properly amended the text to reflect those suggestions. I recommend the editorial team to accept the manuscript in current condition.